# Green Synthesis of Molecularly Imprinted Polymers for Dispersive Magnetic Solid-Phase Extraction of Erythrosine B Associated with Smartphone Detection in Food Samples

**DOI:** 10.3390/ma15217653

**Published:** 2022-10-31

**Authors:** Dounia Elfadil, Flavio Della Pelle, Dario Compagnone, Aziz Amine

**Affiliations:** 1Faculty of Bioscience and Technology for Food, Agriculture and Environment, University of Teramo, via Renato Balzarini 1, 64100 Teramo, Italy; 2Laboratory of Process Engineering and Environment, Faculty of Sciences and Techniques, Hassan II University of Casablanca, Mohammedia 20650, Morocco

**Keywords:** synthetic colorants, food samples, magnetic dispersive solid-phase extraction, polydopamine, molecularly imprinted polymers

## Abstract

Monitoring synthetic colorants in foods is important due to their potential toxicity and pathogenicity. We propose here a new and simple method for the extraction and determination of erythrosine B (ERT-B) in food samples. A composite of polydopamine-based molecularly imprinted polymers coating magnetic nanoparticles (Fe_3_O_4_@PDA@MIP) was synthesized using a green approach and exploited for the magnetic dispersive solid-phase extraction (MDSPE) of ERT-B. Fe_3_O_4_@PDA@MIP provides a rapid extraction of ERT-B, exhibiting good reusability and preconcentration ability. Moreover, the MIP showed a relatively good imprinting factor (3.0 ± 0.05), demonstrating excellent selectivity against patent blue (an interfering dye) and other food matrix components. The proposed MDSPE was coupled to colorimetric smartphone-based detection that allowed us to obtain similar performances of UV–Vis spectroscopy detection. The smartphone-based optical detection facilitated the determination of ERT-B in the 0.5–10 mg/L range, with a limit of detection of 0.04 mg/L. The developed method was successfully employed to determine ERT-B in food samples (juice, candy, and candied cherries) with good recovery values (82–97%).

## 1. Introduction

Erythrosine B (ERT-B) is a popular xanthene coloring agent that is commonly used in foodstuffs, cosmetics, medicines, and textiles [1]. It is highly toxic to humans and can cause many adverse health effects, including cancer and various types of allergies [2,3,4]. Thus, the use of ERT-B in food products is strictly controlled by various international legislations such as the World Health Organization (WHO) and the so-called Codex Alimentarius [5]. The maximum level admitted for ERT-B in candied fruit, chewing gum, and sweet sauces are 200, 50, and 100 mg/kg, respectively (https://www.fao.org/gsfaonline/additives/index.html; accessed on 25 October 2022). Because of the potential hazard to humans, the content of this ‘dye’ in food must be controlled; detection in a simple and fast way is, thus, of major importance.

Different analytical methods such as spectrophotometry [6], Raman spectroscopy [7], high-performance liquid chromatography [8] and electrochemical methods [9,10,11] have been reported for the determination of ERT-B. One of the most important challenges for ERT-B analysis is selectivity, since it is often used together with other colorants, and due to the presence of potential interferent compounds in foods. Thus, regardless of the method used, selective extraction procedures are required for ERT-B analysis in complex matrices to reduce matrix effects and eliminate potential interfering species.

Conventional sample extraction techniques, such as solid-phase extraction (SPE) and liquid–liquid extraction (LLE), have various drawbacks, for example, the need for a high volume of solvents, long and multi-step experimental processes, and the production of a large amount of waste [12,13]. These disadvantages prompt the development of new extraction methods and adsorbents. Different ‘sorbent phases’ have been proposed for ERT-B with their use aimed at bioremediation purposes; among others, hen feathers [14], crop waste pumpkin seed hulls [15], biochar from wood chips and corn cobs [1], chitosan boric acid composite [16], and de-oiled mustard [17] have been employed. Unfortunately, these adsorbent materials are not strictly selective for ERT-B; thus, the development of new effective and selective materials to extract ERT-B from food is still an open challenge. Recently, molecularly imprinted polymers (MIPs) have attracted considerable attention as promising elements for specific recognition in chemical and biological sensors [18]. Molecular imprinting is a technique that generates molecular recognition sites that are chemically and sterically complementary to the target molecule [18,19,20]. In particular, the surface molecular printing technique combined with magnetic nanomaterials has attracted considerable interest because magnetic MIPs can be easily moved/separated by using an external magnet [21]. Recently, dopamine (DA), a biomolecule containing catechol and amine functional groups, has attracted the attention of researchers, particularly for its ability to self-polymerize in an aqueous alkaline or oxidizing solution without crosslinking or initiating agents [22], allowing the realization of different nanomaterial-based devices, sensing systems [23,24,25,26], and MIPs [27,28]. In addition to their use for sample extractions, the other major contribution that has led to improvements in the quantitative application of MIPs regards their combination with non-chromatographic techniques as UV–visible spectroscopy [19], Raman, fluorescence spectroscopy, and electrochemical techniques [20]. Considering the intense color provided by ERT-B in the visible spectrum, it is clearly expected that cost-effective and sensitive measurements with smartphone-based camera technology should lead to successful detection. Thus, taking versatility and simplicity into account, smartphone detection can be efficiently employed in the detection process after the MIP-based extraction of colored complex samples. Therefore, smartphones can be used for tasks usually performed by expensive spectrophotometers, fluorometers, or silicon photodetectors.

The present work reports the development of a novel, rapid and selective method capable of quantifying the ERT-B present in foods. Inspired by the self-polymerization ability of dopamine, the green synthesis of polydopamine-based molecularly imprinted polymer coating magnetic nanoparticles (Fe_3_O_4_@PDA@MIP) was employed; the obtained magnetic MIPs were used to implement a magnetic dispersive solid-phase extraction (MIP-MDSPE) procedure for ERT-B’s selective extraction. The MIP-MDSPE procedure was coupled with straightforward smartphone-based colorimetric detection, which allowed us to guarantee a performance comparable to UV–Vis spectroscopy with the additional advantages of simplicity, rapidity of analysis, and low cost. Eventually, the applicability of the proposed strategy was proven for ERT-B quantification in different foods.

## 2. Materials and Methods

### 2.1. Chemicals and Reagents

Dopamine hydrochloride (≥98%), iron chloride hexahydrate (FeCl_3_⋅6H_2_O), iron chloride tetrahydrate (FeCl_2_·4H_2_O), ammonium hydroxide (NH_4_OH), and ammonium persulfate ((NH_4_)_2_S_2_O_8_ (APS)) (≥98%) were purchased from Sigma Aldrich (Burghausen, Germany). Acetic acid (CH_3_COOH) (≥99.8%) was obtained from Honeywell Fluka (Seelze, Germany). Tris-(hydroxymethyl)-aminomethane buffer (≥99.5%), Erythrosine B (ERT-B) (C_20_H_8_I_4_Na_2_O_5_) (≥80%) and Patent blue V (VF) (C_27_H_31_N_2_NaO_7_S_2_) were purchased from Loba Chemie (Maharashtra, India). All other chemicals used were of analytical grade and were used without further purification.

### 2.2. Apparatus and Image Processing

For smartphone colorimetric analysis, pictures were taken with an android smartphone (Xiaomi Mi 9T, Beijing, China) equipped with a 48-megapixel-resolution (5000 × 3000) camera. The colorimetric detection of ERT-B was conducted by dropping 300 µL of the ERT-B, ERT-B standard, or ERT-B-containing sample onto the ELISA plate microwells. To quantify the intensity of the color, the smartphone camera was used as a signal readout which converted the color intensity of each ELISA well to different RGB color channels. All images were captured manually at an optimized fixed distance from the colorimetric spots in ambient lighting. Each acquisition was performed in triplicate to calculate the standard deviation. Regarding digital analysis, pictures were analyzed using the free open play store application RGB color detector and the color intensities were automatically calculated as different color spaces (RGB, grayscale and weighted value). For absorbance measurements, a Jenway Model 6850 UV–Visible Double-beam Spectrophotometer (Camlab, Cambridge, UK), with 1 cm matched cells was used to perform a comparison with the proposed smartphone method. The absorbance of the pink color of ERT-B was measured at the wavelength of 528 nm against the blank. The characterization with Fourier transform infrared spectroscopy was recorded with an IRAffinity-1S spectrophotometer (SHIMADZU, Kyoto, Japan) in the range of 4000–500 cm^−1^ in an attenuated total reflectance mode.

### 2.3. Preparation of Magnetic Nanoparticles

Magnetic nanoparticles (MNPs) were synthesized by the co-precipitation method [22]. Briefly, 4.3 g of FeCl_2_⋅4H_2_O and 11.68 g of FeCl_3_⋅6H_2_O were added to 200 mL of distilled water. The mixture was stirred at 300 rpm under nitrogen atmosphere. Then, 20 mL of NH_4_OH was added dropwise at 80 °C. The reaction remained under reflux for 40 min and was cooled down at room temperature. The resulting product was then washed with distilled water until a neutral pH was reached. Finally, the MNPs were dried overnight in an oven at 60 °C.

### 2.4. Synthesis of MIPs for ERT-B

The preparation of magnetic-MIPs based on PDA for ERT-B was based on a multistep procedure (Figure 1). Firstly, 20 mg of ERT-B was dissolved in 20 mL of Tris buffer (pH 8.0, 20 mM) containing 100 mg of MNPs and 40 mg of dopamine; then, the mixture was stirred for 12 h, 10 mg of APS was added, and the solution was stirred for 3 h. The obtained products were washed with 0.1 M NaOH. A similar procedure without adding the template was used to prepare the corresponding MNIPs used as control materials.

### 2.5. Binding Experiments

In order to evaluate the binding ability of the synthesized MIPs/NIPs, 2 mg of MIPs/NIPs were suspended in 2 mL acetate buffer solution (0.5 mM, pH 5.0) containing ERT-B from 0.1 to 20 mg/L. After orbital agitation at room temperature (300 rpm) for 10 min, the suspension was separated using an external magnet. UV–Vis spectroscopy was employed to determine the residual concentration of ERT-B at 528 nm.

The equilibrium adsorption capacities of MIPs and NIPs (*Q*, mg/g) were calculated with the following equation:(1)$$Q_{e}=\frac{Ci-Ce}{m}\times V$$
where *Q* is the adsorption capacity at the equilibrium (mg/g), *Ci* and *Ce* are the initial and equilibrium concentrations (mg/L), respectively, *V* is the volume of the adsorption solution (L) and *m* is the weight of the imprinted polymer used (g).

The imprinting factor (*IF*), used to evaluate the specific recognition properties of the synthesized MIPs, is defined as:(2)$$IF=\frac{Q_{MMIP}}{Q_{MNIP}}$$
where *Q* (MIP) (mg/g) and *Q* (NIP) (mg/g) are the adsorption capacities of the MIPs and NIPs, respectively.

The adsorption isotherm was obtained by varying the initial concentration of ERT-B from 0.1 to 20 mg/L. The equilibrium data were fitted using nonlinear Freundlich and Langmuir models. The kinetic experiments were conducted at different times from 1 to 30 min. The experimental kinetic data were extrapolated using nonlinear pseudo-first-order (PFO) and pseudo-second-order (PSO) models. The isotherm and kinetic equations are listed in Appendix A.

The characteristic parameters for each isotherm and kinetic model and the related correlation coefficients have been determined using ORIGIN 8 PRO Software.

### 2.6. MIP-Magnetic Dispersive Solid-Phase Extraction (MIP-MDSPE)

Figure 2 illustrates the dispersive magnetic solid-phase extraction (MIP-MDSPE) procedure; all the experiments were performed in 5 mL polypropylene tubes. First, 4 mg MIPs/NIPs were placed into the tube and conditioned with 2 mL of acetate buffer (0.5 mM, pH 5.0) as adsorption solution; the polymers were magnetically decanted with the aid of a neodymium magnet (40 mm × 20 mm × 10 mm). Then, 2 mL of ERT-B working standard solutions or real sample solutions were added into the tube and orbitally shaken for 10 min at 300 rpm. After removing the supernatant with the aid of a neodymium magnet, 2 mL of 0.1 M NaOH as desorption solvent was added into the tube and orbitally shaken for 10 min at 300 rpm to elute the adsorbed ERT-B. After magnetic separation, the supernatant was collected and analyzed by a UV–Vis spectrophotometer and/or smartphone.

The extraction recoveries were calculated according to the following equation:(3)$$R\%=\frac{Cf_{}}{Ci_{}}\times 100$$
where *Ci* and *Cf* are the initial and final concentrations of ERT-B, respectively.

### 2.7. Determination of ERT-B in Food Samples

Candied cherries, candy, and juice were obtained from a local market. Then, 5 g of sample was dissolved in 15 mL of acetate buffer, sonicated in an ultrasound bath for 5 min, and a warming process (50 °C, 30 min) was used to help the sugar dissolution. The obtained solution was centrifuged, and the supernatant was filtered through a 0.45 μm micropore filter membrane. The filtrate was transferred to a volumetric flask and the volume was adjusted using acetate buffer to reach a final volume of 25 mL.

## 3. Results and Discussion

### 3.1. MIP-Dispersive Solid-Phase Extraction Optimization

To obtain a better interaction between MIPs and ETR-B in an aqueous medium, the effect of pH on ERT-B binding was investigated in the pH 5.0–8.5 range both for MIPs/NIPs. The experimental data (Appendix A) showed that the pH of the solution exerts a strong influence on the adsorption of ERT-B because of the effect on the surface properties of the imprinted materials, as well as on the ionization and dissociation of the dye molecule. The minimum uptake of ERT-B was achieved at pH 8.5 and the maximum adsorption was obtained at pH 5.0. It has been reported that the pKa value for ERT-B is 5.3 [29]; therefore, at pH ≤ 5.3, ERT-B is fully protonated and therefore the selective binding to the MIPs is mainly attributed to hydrogen bonds. In more alkaline solutions, ERT-B is deprotonated, and significant charge repulsion phenomena also probably occur; therefore, the selective binding interactions are reduced. For pH values lower than 5, ERT-B in contact with the adsorbent reacts and the absorbance of the solution decreases largely due to the strong acidity of the medium that leads to ERT-B precipitation [15]. On the other hand, the significant difference in adsorption capacity obtained with the NIPs confirms the action of the complementary cavity of the MIP. In light of the reported observations, pH 5.0 was selected for the following experiments.

The optimal amount of polymer to use in the MIP-MDSPE procedure was found to be 4 mg, since higher amounts do not lead to larger ERT-B retention (Appendix A). Moreover, using 4 mg of polymer, the major difference between MIPs’ and NIPs’ binding capacity was observed. The time required for the maximum adsorption of ERT-B on MIPs was determined by conducting binding experiments from 5 to 30 min; 10 min was chosen as the contact time because longer times did not give rise to significant increases in ERT-B adsorption (Appendix A). The elution time was settled at 10 min since longer times do not lead to increased ERT-B release (Appendix A).

### 3.2. MIPs Performance

#### 3.2.1. Isotherm Modelling

The equilibrium adsorption capacities of MIPs and NIPs for ERT-B were systematically investigated as a function of the initial concentration of ERT-B (Figure 1).

For increasing ERT-B amounts, the uptake by MIPs increased considerably compared with NIPs, indicating that the MIP imprinted cavities that were progressively occupied with ERT-B. Moreover, the excellent reproducibility obtained for all ERT-B concentrations studied (RSD ≤ 5%, *n* = 3) endorses the MIP-MDSPE procedure’s reliability. The imprinting factor (IF) obtained was 3.0 ± 0.05, demonstrating the superior performance of the MIP with respect to NIP. The equilibrium data were fitted using the Langmuir and Freundlich isotherms [30,31,32]; the obtained parameters are summarized in Table 1.

Comparing the R^2^ values obtained, the Langmuir model is the most appropriate for interpreting the adsorption of ERT-B onto the MIPs. The Langmuir isotherm model demonstrates how, for ERT-B, the adsorption process occurs on a monolayered homogeneous surface. Eventually, the MIPs and NIPs exhibited maximum adsorption capacities of 5.79 mg/g and 0.9 m/g, respectively.

#### 3.2.2. Adsorption Kinetics

Appendix A reports the adsorption kinetic data of ERT-B on the MIP and NIP. The adsorption process of ERT-B was very fast and reached a steady state after 10 min. As expected, the MIP exhibited higher adsorption capacities than the NIP since the MIP has imprinted cavities. Two models, including PFO and PSO, were used to analyze the experimental data. The obtained parameters are summarized in Table 1. According to the R^2^, both models fit well with the experimental results. The equilibrium adsorption capacities obtained by PFO were 2.5 and 0.2 mg/g for MIP and NIP, respectively.

#### 3.2.3. FT-IR Characterization

To monitor the MIPs’ fabrication and assess the MIPs’ interactions, FT-IR analyses were carried out for Fe_3_O_4_, Fe_3_O_4_@PDA, ERT-B, MIPs, and NIPs; the results are shown in Figure 2. The absorbance peak that appeared in all FTIR spectra (Figure 2A) at 590 cm^−1^ is attributed to the Fe-O stretching of Fe_3_O_4_ magnetic cores [33,34].

The successful decoration of Fe_3_O_4_ with PDA was confirmed by the presence of N-H bending at around 1585 cm^−1^, C-N stretching at around 1282 cm^−1^ and phenylic C=C stretching at around 1487 cm^−1^ [22]. Given the same composition, similar peaks were also observed for the NIP. In Figure 2B, erythrosine absorption bands at 1602, 1541, and 1455 cm^−1^ are present and were attributed to the benzene ring stretches. The peak at 963 cm^−1^ was assigned to the C=C-H functional group [35,36]. The dominant bands of ERT-B were also observed in the spectra of MIPs after the adsorption step, proving the ERT-B retention; no ERT-B bands were recorded after the elution step, demonstrating the successful removal of ERT-B.

#### 3.2.4. Reusability

The reusability of MIPs has an important role in developing low-cost, reliable, and sustainable analytical applications. Therefore, the reusability test for 10 consecutive adsorption–elution cycles of ERT-B was studied (Figure 3). The mean recovery value obtained for the 10 repetitions was 80%, with RSD values ≤ 4% in all cases. These results indicate that Fe_3_O_4_@PDA@MIP keeps its binding capacity after several uses and could be used up to eight times without significant loss of activity, proving that the as-prepared Fe_3_O_4_@PDA@MIP possessed excellent stability and potential application ability for practical applications.

In this aspect, Fe_3_O_4_@PDA@MIP is used repeatedly after a simple regeneration step involving washing the MIP particles to remove residual analytes and interferents that remain bound after elution. In this way, unlike many other commercially available adsorbent polymers, MIPs are not discarded after a single use. This custom is very important from a green chemistry perspective because it greatly reduces the reagents that might be needed for the repeated synthesis of MIPs for multiple applications.

### 3.3. Analytical Determination of ERT-B

#### 3.3.1. Photometric Detection

Different amounts of ERT-B ranging from 0.5 to 10 mg/L were analyzed with a UV–visible spectrophotometer at 528 nm [37]. Appendix A shows the dose–response curve obtained with the relative equation obtained for the linear range. The calculated limit of detection (LOD) and limit of quantification (LOQ) were 0.03 mg/L and 0.1 mg/L, respectively, as calculated; the LOD and LOQ were calculated according to the formula LOD = 3σ/b and LOQ = 10 × σ/b, where σ is the standard deviation of the intercept and b is the slope of the calibration curve.

#### 3.3.2. Smartphone Detection

Initially, to set the smartphone detection parameters, a series of ERT-B solutions were prepared and placed in ELISA plate microwells; images were acquired with the smartphone using different conditions for the distance of the smartphone (5, 10, 13, and 18 cm), volumes per microwell (50, 100, 200, 300, and 400 mL), and lighting (flashlights on/off). The color intensities were automatically calculated as different color spaces; the RGB color detector application (available app on smartphones) was used to directly determine the color channels with the smartphone. Thus, color intensities were plotted as a function of ERT-B concentrations, as shown in Appendix A. The measurement of color intensities with smartphones allowed us to plot the intensity of red, green, and blue color spaces as a function of ERT-B concentration. The green color space, as expected, showed a strong decrease with increasing ERT-B concentration, proving higher sensitivity, and was selected as an optimal quantitative analytical parameter among all tested color spaces. To investigate the best imaging conditions, the distance between the microwell and the smartphone camera was optimized by measuring the intensity of the green channel (Appendix A); distances ranging from 5 to 18 cm were tested. A large variation in green color intensity was observed for all distances studied; 13 cm was chosen, since it allowed us to obtain higher color intensity compared to 10 and 18 cm. Regarding the lighting for the image acquisition, they were acquired under daylight and in a dark box using an external light (flashlights on); the images in daylight provide greater sensitivity, and thus this mode was selected. The ERT-B volume per well was tested from 50 µL to 400 µL; 300 µL gave higher sensitivity. In summary, the best conditions selected were: microwell volume of 300 µL, focal distance of 13 cm, daylight acquisition, and data interpretation using the green channel.

The ERT-B calibration curve obtained with the smartphone following the green color channel is presented in Figure 4.

The results showed a linear range from 0.5 to 10 mg/L, with a determination coefficient of R^2^ = 0.9920. The LOD and LOQ were 0.04 and 0.13 mg/L, respectively. The comparison between the UV–Vis spectrophotometer and smartphone detection showed no significant difference. However, the measurement with the smartphone is rapid, low cost, and allows in-field analysis without causing a loss of sensitivity.

### 3.4. Selectivity Study

The MIP selectivity was tested toward different compounds, in particular, mixtures of ERT-B in the presence of patent blue (5 mg/L), ascorbic acid (10 mg/L), glucose (10 mg/L), fructose (10 mg/L), sucrose (10 mg/L), benzoic acid (5 mg/L), magnesium chloride (5 mg/L), potassium nitrate (5 mg/L), sodium sulfate (5 mg/L) and zinc chloride (5 mg/L). The obtained results are reported in Figure 5.

The MIP showed low adsorption capacities toward the patent blue dye. The presence of interfering ions in the ERT-B solution induced a slight adsorption capacity change compared with that of pure ERT-B solution (less than 5% change) demonstrating a good selectivity of the MIP. This is due to the large number of imprinted cavities in Fe_3_O_4_@PDA@MIP which have a strong chemical and structural affinity for ERT-B. These cavities allow ERT-B to enter the polymer system with high affinity.

### 3.5. Preconcentration

MDSPE is commonly used as a preconcentration or separation technique when complex matrices or low concentrations of analytes need to be analyzed [38]. For real sample analysis using preconcentration, the sample volume and elution volume are the most important parameters to achieve high preconcentration factors. Therefore, in this study, the effect of elution volume on the recovery of analyte extraction was examined; the recovery values as a function of sample elution volume are presented in Figure 6.

The preconcentration factor was calculated using the ratio of the sample charged volume (2 mL) on the elution volume (2, 1, 0.5, 0.4, 0.33 mL); up to a concentration factor of 5 times, there are no effective losses of MIP. Employing the preconcentration factor ‘5’, a theoretical LOD of 8 µg/L could be reached.

### 3.6. Real Samples

To verify the exploitability of the established method for real sample analysis, it was applied for the analysis of ERT-B in juice, candied cherries and candy. The samples were spiked at three ERT-B levels, extracted using the optimized MIP-MDSPE procedure, and analyzed in parallel with the photometric and smartphone-based methods. The data obtained are reported in Table 2, while pictures and the absorption spectra of the fortified samples extracted using the MIP-MDSPE procedure are reported in Appendix A.

The data obtained with both methods demonstrate that the MIP-MDSPE procedure is suitable for real sample analysis. The smartphone-based method shows ERT-B quantification and recoveries (82–96%) similar to the photometric determination (80–93); RSD ≤ 6% proves the proposed method’s reproducibility.

To the best of our knowledge, this is the first report that deals with the combination of sample clean-up and the smartphone determination of erythrosine B in real food samples. Compared with the existing methods for the determination of ERT-B in real samples (Table 3), the main advantages of the proposed method are: (i) the process of formation of the MIP is green and requires few chemicals, (ii) the strategy without preconcentration procedures allows us to measure the ERT-B at the law limits, (iii) and smartphone-based colorimetric detection allows simple, rapid, and low-cost ERT-B analysis. Eventually, the applicability of the proposed strategy was proved for ERT-B quantification in different foods with a good recovery. (iv) The main advantage of our strategy remains in the combination of sample clean-up using MIP and the fast and affordable readout method for on-site food safety monitoring.

## 4. Conclusions

In this work, a novel magnetic MIP based on PDA was developed for the selective, colorimetric determination of ERT-B; the Fe_3_O_4_@PDA@MIP has been synthesized following a sustainable strategy. An MDSPE extraction procedure for ERT-B based on the MIP was optimized and coupled with straightforward smartphone-based detection. The extraction procedure is effective and reproducible, while smartphone detection enables sensitive and repeatable ERT-B quantification. The smartphone-based detection performance was found to be similar to the conventional photometric-based method, which requires the use of a conventional spectrophotometer.

Summing up, an effective MIP-MDSPE procedure has been proposed for ERT-B extraction from real samples. In addition, the proposed ERT-B determination strategy via smartphone induces affordable, simple, and rapid analysis.

## Data Availability

Not applicable.

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
