# Peer review of "Green Synthesis of Molecularly Imprinted Polymers for Dispersive Magnetic Solid-Phase Extraction of Erythrosine B Associated with Smartphone Detection in Food Samples"

_materials, 2022, doi:10.3390/ma15217653_

Round 1

Reviewer 1 Report

This work presented a method for preparation of molecularly imprinted polymers as sorbent of dispersive magnetic solid phase extraction for precontraction of erythrosine B. Sample pretreatment as a crucial step for the analysis of trace compounds in complex samples received considerable attention in recently. In generally, this is an interesting work, but some of contents have not well organized. So, before it can be accepted for publication, the following comments should be well addressed.

1)      The first paragraph of introduction can be divided into three parts. The first part is introduction of Erythrosine-B. The sample pretreatment and analytical method can be presented as second part. The third part is introduction of the advantages of dispersive magnetic solid phase extraction

2)      Please unify the erythrosine B(title) and erythrosine-B (abstract and text)

3)      Lines 48-51, the content of this sentence is very confused here, I don’t know why authors emphasize bioremediation materials. I suggestion that the authors can list some star materials of sample pretreatment, such as graphene (Talanta 2020, 208, 120440), carbon nitrite (Chin. Chem. Lett. 2022, 33, 903-906) and MOFs (J. Hazard. Mater. 2022, 424, 127559; Talanta 2021, 235, 122818)

4)      Lines 86-88 and 113-114, the authors should provide correct chemical formular

5)      The desorption parameters (conditions) including desorption time and desorption solvents should be provided in the text

6)      SEM characterization of the prepared sorbent should be provided

7)      Line 279, a blank should be added between number and unit for 50μL, 400μL and 300μL

8)      How to calculate the LOD and LOQ should be provided in the text

9)      Line 293, a blank should be added between number and unit for Ascorbic acid (10mg/L)

10)  Table 2, for line RSD, please check 4%, 2%... is correct or not? Because a “%” is presented following RSD

Author Response

Reviewer 1

This work presented a method for preparation of molecularly imprinted polymers as sorbent of dispersive magnetic solid phase extraction for precontraction of erythrosine B. Sample pretreatment as a crucial step for the analysis of trace compounds in complex samples received considerable attention in recently. In generally, this is an interesting work, but some of contents have not well organized. So, before it can be accepted for publication, the following comments should be well addressed.

We are honored that the reviewer liked the work

1) The first paragraph of introduction can be divided into three parts. The first part is introduction of Erythrosine-B. The sample pretreatment and analytical method can be presented as second part. The third part is introduction of the advantages of dispersive magnetic solid phase extraction

The introduction has been organized as suggested by the reviewer

2) Please unify the erythrosine B (title) and erythrosine-B (abstract and text)

      The way of writing erythrosine was standardized throughout the entire paper

3) Lines 48-51, the content of this sentence is very confused here, I don’t know why authors emphasize bioremediation materials. I suggestion that the authors can list some star materials of sample pretreatment, such as graphene (Talanta 2020, 208, 120440), carbon nitrite (Chin. Chem. Lett. 2022, 33, 903-906) and MOFs (J. Hazard. Mater. 2022, 424, 127559; Talanta 2021, 235, 122818)

       We thank the reviewer for the suggestion, but the materials cited in the manuscript were used for erythrosine B adsorption. However, the suggested references were not used for ERT-B adsorption, so we did not include them in the references.

4) Lines 86-88 and 113-114, the authors should provide correct chemical formular

    In the suggested lines the correct chemical formula has been reported

5) The desorption parameters (conditions) including desorption time and desorption solvents should be provided in the text

To better explain the 'desorption time' and 'desorption solvents' this discussion has been added in section 2.6 in the revised manuscript

Scheme 2 illustrates the dispersive magnetic solid-phase extraction (MIP-MDSPE) procedure; all the experiments were performed in 5 mL polypropylene tubes. 4 mg MIPs/NIPs were placed into the tube and conditioned with 2 mL of acetate buffer (0.5mM, pH 5.0) as adsorption solution, the polymers were magnetically decanted with the aid of a neodymium magnet (40mm x 20mm x 10mm). Then, 2 mL of ERT-B working standard solutions or real sample solutions were added into the tube and orbitally shaken for 10min at 300 rpm. After removing the supernatant with the aid of a neodymium magnet, 2 mL of 0.1 M NaOH as desorption solvent was added into the tube and orbitally shaken for 10min at 300 rpm to elute the adsorbed ERT-B. After magnetic separation, the supernatant was collected and analyzed by UV–Vis spectrophotometer and/or smartphone.

6)      SEM characterization of the prepared sorbent should be provided

       We thank the reviewer for the comment.

        Concerning SEM, the Fe3O4@PDA synthesis protocol was performed in accordance with other previous papers that have already fully characterized these materials by SEM and other characterization techniques [3] [10-12]. We have preferred to perform the characterization with FT-IR for the Fe3O4, Fe3O4@PDA and Fe3O4@PDA@MIP, not only to highlight the materials' features but also to prove the MIP effectiveness (see Figure 2).

Figure 2. A) FT-IR spectra acquired for Fe3O4, Fe3O4@PDA@NIP and Fe3O4@PDA@MIP; (B) FT-IR spectra acquired for ERT-B, Fe3O4@PDA@MIP before ERT-B adsorption, Fe3O4@PDA@MIP after ERT-B adsorption, Fe3O4@PDA@MIP after ERT-B elution.

7) Line 279, a blank should be added between number and unit for 50μL, 400μL and 300Μl

Thanks for the suggestion, the mistake has been corrected

8) How to calculate the LOD and LOQ should be provided in the text

The way to calculate the LOD and LOQ has been reported in the text as reported below and in section 3.3.1.

       The LOD and LOQ was calculated according to the formula LOD = 3σ/b and LOQ=10 X σ/b, where σ is the standard deviation of the intercept and b the slope of the calibration curve.

9) Line 293, a blank should be added between number and unit for Ascorbic acid (10mg/L)

Thanks for the suggestion, the mistake has been corrected

10)  Table 2, for line RSD, please check 4%, 2%... is correct or not? Because a “%” is presented following RSD

The percentage close to the RSD values has been removed.

Reviewer 2 Report

The manuscript by A. Amine and coworkers reports the magnetic nanoparticles for molecular imprinted of Erythrosine-B (ERT-B) and its analysis in food samples. The manuscript is interesting and it can be considered for publication; I have only a few points to pose to the author's attention:

-It is not clear what are the advantages (besides selectivity) of using this method compared to other similar ones. I suggest to add a comparative table of methods for ERT-B detection in real samples.

-No information is reported on the stability of the nanoparticles. Please add this information. Please, add some information about it.

-The authors developed a smartphone method for ERT-B analysis. This is very interesting in order to do in situ analysis. How could the in situ analysis be carried out?

Please check text editiong:

in subscripts and superscript such as "FeCl3· 6H2O" and  "cm-1".

line 317  "elution volume (2, 1, 0.5, 0.4, 0.33 mL)".

line 318 preconcentration factor 5'

Author Response

Response to reviewers

Article

Green synthesis of molecularly imprinted polymers for dispersive magnetic solid phase extraction of erythrosine B associated with smartphone detection in food samples

Dounia Elfadil1,2, Flavio Della Pelle1, Dario Compagnone 1,* and Aziz Amine 2,*

Reviewer

The manuscript by A. Amine and coworkers reports the magnetic nanoparticles for molecular imprinted of Erythrosine-B (ERT-B) and its analysis in food samples. The manuscript is interesting and it can be considered for publication; I have only a few points to pose to the author's attention:

1) It is not clear what are the advantages (besides selectivity) of using this method compared to other similar ones. I suggest to add a comparative table of methods for ERT-B detection in real samples.

We thank the reviewer for the suggestions to improve the paper.

This discussion and table have been added to the main text of the paper in section 3.6

To the best of our knowledge, this is the first report that deals with the combination of sample clean up using a plastic antibody (MIP) and smartphone determination of Erythrosine B in real food samples. Compared to the existing methods for the determination of ERT-B in real samples (Table 3), the main advantage of the proposed method lies in the fact that, (i) The smartphone-based colorimetric detection, allowed simple, rapide, and low cost ERT-B analysis.  Eventually, the applicability of the proposed strategy was proved for ERT-B quantification in different foods with a good recovery (ii) the process of formation of the MIP is green and requires few chemicals, (iii) the strategy without pre-concentration procedures allows to measure the ERT-B at the law limits, (iv) the main advantage of our strategy remains in the combination of sample cleanup by using MIP and the fast and affordable readout method for on-site food safety monitoring.

  Table 3: comparative table of methods for ERT-B detection in real food samples

Method

LOD (mg/L)

Real sample

Recovery

Ref

ELISA

0.0022

Healthy energy drink, breezer, grape juice, coca-cola sugar, fermented bean curd and tomato paste

86.3% to 115.5%

[1]

Micellar electrokinetic capillary chromatography

0.01

Soft Drinks and Confectioneries

82%

[2]

Capillary zone electrophoresis

0.35

Bitter, Grenadine, Ice lolly, strawberry,and raspberry.

111% to 95.00%

[3]

High-performance liquid chromatography using a short column with photodiode array detection

0.011

Soft drinks and candies

76.6 to 115.0%

[4]

HPLC-DAD method

0.026

Wine red dry, Milk shake, Fish, Tomato,and paste.

83.7 to 107.5%

[5]

Colorimetric reading via smartphone

0.04

Juice, candy, and candied cherries

82% to 97%

This work

2) No information is reported on the stability of the nanoparticles. Please add this information. Please, add some information about it.

We thank the reviewer for the comment

Fe3O4 nanoparticles are known by their long stability as reported by many researchers [6-10],Furthermore, Since the nanoparticles are decorated with dopamine polymer, they are therefore not accessible to be altered by the medium effect, Moreover, a reusability study has been done and demonstrated that the Fe3O4@PDA@MIP are reusable for 8 times without any significant loss (RSD values ≤4%) of its activity (Figure 3).

Figure 3. Reusability study of MIP-MDSPE (ERT-B 1 mg/L).

3) The authors developed a smartphone method for ERT-B analysis. This is very interesting in order to do in situ analysis. How could the in situ analysis be carried out?

For the in situ analysis of ERT-B using the developed MIP-based dispersive solid-phase extraction protocol combined with smartphone readout, no instruments were required. In fact, common laboratory tools simply involve a magnet for separation and a smartphone as the readout instrument, making this approach potentially feasible anywhere.

4) Please check text editing: in subscripts and superscript such as "FeCl3· 6H2O" and "cm-1". line 317 "elution volume (2, 1, 0.5, 0.4, 0.33 mL)". line 318 preconcentration factor 5

Subscripts and superscripts have been edited and corrected throughout the entire manuscript.

Reviewer 3 Report

The current work focuses on the Green synthesis of molecularly imprinted polymers for dispersive magnetic solid phase extraction of erythrosine B associated with smartphone detection in food samples. The author’s great effort into the improved manuscript, but minor issues should be addressed. 

Even if the manuscript is based on the method capable of quantifying ERT-B present in foods. Full characterization of the main material is important and required, as it determines the properties of the used material.

DLS and Zeta potential for detecting its particle size distribution and stability

XRD and TEM for the clear type of magnetic phase and particle size

VSM to measure its magnetic efficacy which is important for activity and reusing

Abstract

- Correct typo(Fe3O4@PDA@MIP) to (Fe3O4@PDA@MIP)

- (82%-97%) to (82-97%)

Introduction

- The introduction doesn’t provide sufficient background, and all relevant references are not included. 

- The novelty of this work is not highlighted and the author's contribution was not clear compared to other previous works. 

-What is the minimum determination limit of ERT-B by previous work to the current work? Speed of measurement? Accuracy? recovery?

Materials and Methods 

- The used reagents with their impurities should be inserted

- Experimental part required rephrasing to be more precise with details and logic to the reader for reproducibility. Especially Binding experiments section & MIP- magnetic dispersive solid phase extraction section 

- Correct typo FeCl2⋅4H2O, FeCl3⋅6H2O, NH4OH,….

- Preparation of magnetic nanoparticles: not clear whether they use a stirrer for mixing or not. Type? Speed? Using an open system? or closing under nitrogen?

 Results and discussion 

One of the main problems in the manuscript is that the authors show only results without any interpretations of it or confirmation by citation. More details are required to explain the obtained results e.g.

MIPs performance section and Analytical determination of ERT-B section 

- Figures should be clear e.g. Fig 4, S5, need to redraw, and also sample photo should come with high resolution

- Also, it should clear the author's contribution in comparison to other previous works during the discussion. Supported table with a comparison with previous work based on the material used, minimum determination limit? efficiency and recoveries of ERT-B,....

Author Response

Response to reviewer 3

Article

Green synthesis of molecularly imprinted polymers for dispersive magnetic solid phase extraction of erythrosine B associated with smartphone detection in food samples

Dounia Elfadil1,2, Flavio Della Pelle1, Dario Compagnone 1,* and Aziz Amine 2,*

Reviewer 3

1) The current work focuses on the Green synthesis of molecularly imprinted polymers for dispersive magnetic 1solid phase extraction of erythrosine B associated with smartphone detection in food samples. The author’s great effort into the improved manuscript, but minor issues should be addressed. Even if the manuscript is based on the method capable of quantifying ERT-B present in foods. Full characterization of the main material is important and required, as it determines the properties of the used material.

DLS and Zeta potential for detecting its particle size distribution and stability

XRD and TEM for the clear type of magnetic phase and particle size

VSM to measure its magnetic efficacy which is important for activity and reusing

We greatly value the reviewer's comment which we found very useful.

Regarding the characterization strategies of our material, the synthesis protocol of Fe3O4@PDA was performed in accordance with previous papers that have already characterized these materials by SEM, XRD, DLS and other characterization techniques[3][1-3]. Since the synthesis of Fe3O4@PDA was carried out in accordance with previous work in the literature, we preferred to perform FT-IR characterization for Fe3O4, Fe3O4@PDA and Fe3O4@PDA@MIP, not only to demonstrate the characteristics of the materials but also to prove the effectiveness of the MIP (see Figure 2).In addition the study of the adsorption capacity (see Figure 1) indirectly demonstrates the success of the MIP synthesis.

Figure 1. Binding isotherms obtained with MIPs and NIPs for ERT-B.

Figure 2. A) FT-IR spectra acquired for Fe3O4, Fe3O4@PDA@NIP and Fe3O4@PDA@MIP; (B) FT-IR spectra acquired for ERT-B, Fe3O4@PDA@MIP before ERT-B adsorption, Fe3O4@PDA@MIP after ERT-B adsorption, Fe3O4@PDA@MIP after ERT-B elution.

2) Abstract

- Correct typo (Fe3O4@PDA@MIP) to (Fe3O4@PDA@MIP)

- (82%-97%) to (82-97%)

We have tried to correct all errors and oversights, not only in the abstract but throughout the paper.

2) Introduction

- The introduction doesn’t provide sufficient background, and all relevant references are not included. 

The introduction has been improved, also adding relevant and significant references.

3) The novelty of this work is not highlighted and the author's contribution was not clear compared to other previous works. 

We thank the reviewer for helping us to improve the paper's novelty in the introduction.

The present work reports the development of a novel, rapid and selective method capable of quantifying ERT-B present in foods. Inspired by the self-polymerization ability of dopamine, green synthesis of polydopamine-based molecularly imprinted polymer coating magnetic nanoparticles (Fe3O4@PDA@MIP) was employed; the obtained magnetic MIPs were used to implement a magnetic dispersive solid phase extraction (MIP-MDSPE) procedure for the ERT-B selective extraction. The MIP-MDSPE was coupled to a straightforward smartphone-based colorimetric detection, which allowed to guarantee performance comparable to UV-Vis spectroscopy. With the additional advantages of simplicity, rapidity of analysis, and low cost.  Eventually, the applicability of the proposed strategy was proved for ERT-B quantification in different foods

To this aim, this discussion and table have been added to the main text of the paper in section 3.6

To the best of our knowledge, this is the first report that deals with the combination between sample clean-up and smartphone determination of Erythrosine B in real food samples. Compared to the existing methods for the determination of ERT-B in real samples (Table 3), the main advantage of the proposed method lies in the fact that (i) the process of formation of the MIP is green and requires few chemicals, (ii) the strategy without pre-concentration procedures allows to measure the ERT-B at the law limits, (iii) The smartphone-based colorimetric detection, allowed simple, rapid, and low-cost ERT-B analysis. Eventually, the applicability of the proposed strategy was proved for ERT-B quantification in different foods with a good recovery, (vi) the main advantage of our strategy remains in the combination of sample cleanup by using MIP and the fast and affordable readout method for on-site food safety monitoring.

4) What is the minimum determination limit of ERT-B by previous work to the current work? Speed of measurement? Accuracy? recovery?

A comparative study with other previous work that determined ERT-B in real samples was performed, the minimum determination (LOD) and other analytical performances such as recovery were listed and discussed in Table 3.

This discussion and table have been added to the main text of the paper in section 3.6

To the best of our knowledge, this is the first report that deals with the combination between sample clean-up and smartphone determination of Erythrosine B in real food samples. Compared to the existing methods for the determination of ERT-B in real samples (Table 3), the main advantage of the proposed method lies in the fact that (i) the process of formation of the MIP is green and requires few chemicals, (ii) the strategy without pre-concentration procedures allows to measure the ERT-B at the law limits, (iii) The smartphone-based colorimetric detection, allowed simple, rapid, and low-cost ERT-B analysis. Eventually, the applicability of the proposed strategy was proved for ERT-B quantification in different foods with a good recovery, (vi) the main advantage of our strategy remains in the combination of sample cleanup by using MIP and the fast and affordable readout method for on-site food safety monitoring.

  Table 3: comparative table of methods for ERT-B detection in real food samples

Method

LOD (mg/L)

Real sample

Recovery

Ref

ELISA

0.0022

Healthy energy drink, breezer, grape juice, coca-cola sugar, fermented bean curd and tomato paste

86.3% to 115.5%

[4]

Micellar electrokinetic capillary chromatography

0.01

Soft Drinks and Confectioneries

82%

[5]

Capillary zone electrophoresis

0.35

Bitter, Grenadine, Ice lolly, strawberry,and raspberry.

111% to 95.00%

[6]

High-performance liquid chromatography using a short column with photodiode array detection

0.011

Soft drinks and candies

76.6 to 115.0%

[7]

HPLC-DAD method

0.026

Wine red dry, Milk shake, Fish, Tomato,and paste.

83.7 to 107.5%

[8]

Colorimetric reading via smartphone

0.04

Juice, candy, and candied cherries

82% to 97%

This work

5) Materials and Methods 

- The used reagents with their impurities should be inserted

The purity of the reagents has been added in section  2.1. Chemicals and reagents

2.1. Chemicals and reagents

Dopamine hydrochloride (≥98%), iron chloride hexahydrate (FeCl3⋅6H2O), iron chloride tetrahydrate (FeCl2⋅4H2O), ammonium hydroxide (NH4OH), and ammonium persulfate ((NH4)2S2O8 (APS))(≥98%) were purchased from Sigma Aldrich (Germany). Acetic acid (CH3COOH) (≥99.8%) was obtained from Honeywell Fluka (Germany). Tris-(hydroxymethyl)-aminomethane buffer (≥99.5%), Erythrosine B (ERT-B)(C20H8I4Na2O5) (≥80%)  and Patent blue V (VF) (C27H31N2NaO7S2) were purchased from Loba chemie (India). All other chemicals used were of analytical grade and were used without further purification. 

6) Experimental part required rephrasing to be more precise with details and logic to the reader for reproducibility. Especially Binding experiments section & MIP- magnetic dispersive solid phase extraction section 

We thank the reviewer for the useful comment. The Experimental part has been improved.

7)  Correct typo FeCl2⋅4H2O, FeCl3⋅6H2O, NH4OH,….

We have tried to correct all errors and oversights, not only in the abstract but throughout the paper.

8)  Preparation of magnetic nanoparticles: not clear whether they use a stirrer for mixing or not. Type? Speed? Using an open system? or closing under nitrogen?

The synthesis strategy description has been improved, as reported below in section 2.3 of the revised manuscript

Magnetic nanoparticles (MNPs) were synthesized by the co-precipitation method [2]. Briefly, 4.3 g of FeCl2⋅4H2O and 11.68 g of FeCl3⋅6H2O were added to 200 mL of distilled water. The mixture was stirred at 300 rpm under nitrogen atmosphere. Then, 20 mL of NH4OH was added dropwise at 80 °C. The reaction remained under reflux for 40 min and was cooled down at room temperature. The resulting product was then washed with distilled water until neutral pH was reached. Finally, the MNPs were dried overnight in an oven at 60 °C.

9) Results and discussion 

One of the main problems in the manuscript is that the authors show only results without any interpretations of it or confirmation by citation. More details are required to explain the obtained results e.g.

The discussion of the results has been improved to make it more effective.

  1. MIPs performance section and Analytical determination of ERT-B section 

- Figures should be clear e.g. Fig 4, S5, need to redraw, and also sample photo should come with high resolution

The required Figures have been improved and all the paper has been improved to make it more effective.

10) Also, it should clear the author's contribution in comparison to other previous works during the discussion. Supported table with a comparison with previous work based on the material used, minimum determination limit? efficiency and recoveries of ERT-B,....

We thank the editor for the suggestions to improve the paper.

This discussion and table have been added to the main text of the paper in section 3.6

To the best of our knowledge, this is the first report that deals with the combination between sample clean-up and smartphone determination of Erythrosine B in real food samples. Compared to the existing methods for the determination of ERT-B in real samples (Table 3), the main advantage of the proposed method lies in the fact that (i) the process of formation of the MIP is green and requires few chemicals, (ii) the strategy without pre-concentration procedures allows to measure the ERT-B at the law limits, (iii) The smartphone-based colorimetric detection, allowed simple, rapid, and low-cost ERT-B analysis. Eventually, the applicability of the proposed strategy was proved for ERT-B quantification in different foods with a good recovery, (iv) the main advantage of our strategy remains in the combination of sample cleanup by using MIP and the fast and affordable readout method for on-site food safety monitoring.

  Table 3: comparative table of methods for ERT-B detection in real food samples

Method

LOD (mg/L)

Real sample

Recovery

Ref

ELISA

0.0022

Healthy energy drink, breezer, grape juice, coca-cola sugar, fermented bean curd and tomato paste

86.3% to 115.5%

[4]

Micellar electrokinetic capillary chromatography

0.01

Soft Drinks and Confectioneries

82%

[5]

Capillary zone electrophoresis

0.35

Bitter, Grenadine, Ice lolly, strawberry,and raspberry.

111% to 95.00%

[6]

High-performance liquid chromatography using a short column with photodiode array detection

0.011

Soft drinks and candies

76.6 to 115.0%

[7]

HPLC-DAD method

0.026

Wine red dry, Milk shake, Fish, Tomato,and paste.

83.7 to 107.5%

[8]

Colorimetric reading via smartphone

0.04

Juice, candy, and candied cherries

82% to 97%

This work

References

  1. Luo, M.; Zhang, Y.; Zhao, S. Non-Enzymatic Hydrogen Peroxide Sensor Based on Fe3O4@Polydopamine-Ag Nanocomposite Modified Magnetic Glassy Carbon Electrode. Journal of The Electrochemical Society 2021, 168, 067511, doi:10.1149/1945-7111/ac0604.
  2. Lamaoui, A.; Palacios-Santander, J.M.; Amine, A.; Cubillana-Aguilera, L. Molecularly imprinted polymers based on polydopamine: Assessment of non-specific adsorption. Microchemical Journal 2021, 164, 106043.
  3. Liu, C.; Zhou, Y.; Wu, G.; Gao, K.; Li, L.; Tu, H.; Chen, Z. Sandwich-likely structured, magnetically-driven recovery, biomimetic composite penicillin G acylase-based biocatalyst with excellent operation stability. Colloids and Surfaces A: Physicochemical and Engineering Aspects 2022, 639, 128245, doi:https://doi.org/10.1016/j.colsurfa.2021.128245.
  4. Xu, Z.; Zheng, L.; Yin, Y.; Wang, J.; Wang, P.; Ren, L.; Eremin, S.A.; He, X.; Meng, M.; Xi, R. A sensitive competitive enzyme immunoassay for detection of erythrosine in foodstuffs. Food Control 2015, 47, 472-477, doi:https://doi.org/10.1016/j.foodcont.2014.07.033.
  5. Chou, S.-S.; Lin, Y.-H.; Cheng, C.-C.; Hwang, D.-F. Determination of Synthetic Colors in Soft Drinks and Confectioneries by Micellar Electrokinetic Capillary Chromatography. Journal of Food Science 2002, 67, 1314-1318, doi:https://doi.org/10.1111/j.1365-2621.2002.tb10280.x.
  6. Berzas Nevado, J.J.; Guiberteau Cabanillas, C.; Contento Salcedo, A.M. Method development and validation for the simultaneous determination of dyes in foodstuffs by capillary zone electrophoresis. Analytica Chimica Acta 1999, 378, 63-71, doi:https://doi.org/10.1016/S0003-2670(98)00574-1.
  7. Yoshioka, N.; Ichihashi, K. Determination of 40 synthetic food colors in drinks and candies by high-performance liquid chromatography using a short column with photodiode array detection. Talanta 2008, 74, 1408-1413, doi:https://doi.org/10.1016/j.talanta.2007.09.015.
  8. Palianskikh, A.I.; Sychik, S.I.; Leschev, S.M.; Pliashak, Y.M.; Fiodarava, T.A.; Belyshava, L.L. Development and validation of the HPLC-DAD method for the quantification of 16 synthetic dyes in various foods and the use of liquid anion exchange extraction for qualitative expression determination. Food Chemistry 2022, 369, 130947, doi:https://doi.org/10.1016/j.foodchem.2021.130947.
